# SUPA-EEG : Scale-Unified Parieto-occipital Architecture for EEG

**Finn Beckmann** — 2025403378,
finn.beckmann@online.de,

**Natthanan Bhukan** — 2025280434
panyx25@mails.tsinghua.edu.cn

**Huy Hai Nguyen** — 2025280353,
rhh25@mails.tsinghua.edu.cn,

**Lydia Liya He** — 2026403078
l32he@uwaterloo.ca

## 1 Introduction

Electroencephalography (EEG) is a non-invasive spatio-temporal recording of cerebral cortical activity [7]. Decoding visual perception by concurrent EEG-probes offers a path towards applications including brain-computer interfaces, restorative prosthetics, and cognitive-load monitoring. The recently released THINGS-EEG dataset [4] pairs 64-channel EEG with large-scale image stimuli, enabling fine- and coarse-grained visual decoding at scale. However, EEG signals are noisy, high-dimensional, and exhibit strong inter-subject variability, causing standard supervised pipelines and generalize poorly.

Despite recent progress, three fundamental limitations persist across existing methods. First, all current approaches including NICE [10], ATM [9], and CognitionCapturer [11] align EEG representations to the final layer of CLIP, which undergoes representation collapse due to augmentation-based contrastive learning, discarding the low-level and mid-level visual structure that neural signals are known to preserve. Second, all existing methods treat all EEG channels equally, despite visual perception being primarily reflected in occipital and parieto-occipital electrodes, while frontal and central channels capture unrelated cognitive processes introducing noise that obscures the visual signal during learning. Third, no existing method accounts for the multi-scale nature of visual processing, in which the brain encodes low-level edges and colors, mid-level spatial structure, and high-level object semantics progressively across cortical regions, yet all methods compress this rich hierarchy into a single flat EEG embedding.

### 1.1 Proposed Method

We propose **SUPA-EEG**, Scale-Unified Parieto-occipital Architecture for EEG for zero-shot visual retrieval on THINGS-EEG. SUPA-EEG addresses all three limitations through a unified scale conditioning vector, which serves three roles simultaneously. First, it performs soft channel selection by learning to emphasise visually relevant occipital and parieto-occipital electrodes at fine scales and a broader cortical network at coarser scales. Next, it conditions a shared LeWM-style transformer encoder to extract scale-specific spatiotemporal representations from multi-scale patch tokens of increasing receptive field such as fine, medium, and coarse. Last, it identifies the corresponding I-JEPA visual scale target for alignment. The resulting scale-specific EEG embeddings are aligned to the corresponding intermediate features of a frozen I-JEPA encoder via scale-wise InfoNCE, stabilised by SIGReg to prevent representation collapse on small EEG datasets, and regularised with sparsity constraints to encourage biologically interpretable channel selection.

## 2 Related Works

### 2.1 Visual Stimulus EEG Datasets

A main benchmark for EEG-based visual decoding is the THINGS-EEG dataset [4], which provides EEG recordings paired with images drawn from the THINGS object concept database, enabling systematic evaluation of neural decoding across a broad range of visual categories.

## 2.2 EEG Encoding Architectures

Early decoding methods approached this problem through discriminative classification. EEGNet [8] introduced a compact depthwise separable Conv2D architecture that has since become the canonical baseline for raw-window EEG classification. RGNN [12] further incorporates a graph neural network over EEG channels, leveraging the known electrode topology of the scalp montage to model spatial dependencies. ATM [9] advances this direction by modeling temporal dynamics through attention mechanisms, treating decoding as a direct classification problem over visual categories without explicitly grounding representations in semantic space. Also, BraVL [3] aligns EEG with both visual and linguistic embeddings through variational learning, while NICE [10] employs contrastive learning to match EEG and image representations. CognitionCapturer [11] further advances this direction by learning richer semantic representations grounded in both vision and language, enabling more expressive retrieval-based decoding from neural signals.

## 2.3 Foundation Models for Time Series

LLMFew [2] demonstrates that Transformer architectures pretrained on language can transfer to time-series classification in the few-shot regime, outperforming state-of-the-art baselines across 10 UEA datasets [1] by significant margins, suggesting that large pretrained models carry useful inductive biases for structured temporal signals such as EEG. Motivated by the same insight S-JEPA [5] adapts this principle directly to EEG by introducing a spatial block masking strategy over electrode channels, providing preliminary evidence that JEPA-based pretraining can enable seamless cross-dataset transfer in EEG signal processing.

## 3 Challenges

The core difficulties of this project can be separated into two main aspects, which are the inherent properties of the EEG signal and the complexity of aligning it to visual representations.

First, EEG is an inherently noisy signal in which class-discriminative neural components occupy only a small fraction of the total variance, while ocular, muscular, and line-noise artifacts dominate. Inter-subject variability Inter-subject variability, stemming from variations in electrode placement and individual brain anatomy, exacerbates this difficulty. Additionally, differences among participants result in a model learned from one subject not transferring well to another, which creates a persistent distribution shift that limits generalization.

Second, building a model that connects the EEG and visual modalities across multiple scales introduces several non-trivial design challenges. Scales introduce several non-trivial design challenges. The multi-scale nature of I-JEPA (Image Joint Embedding Predictive Architecture) or Transformer, producing local, spatial, and semantic feature targets at different granularities, requires a corresponding multi-scale EEG encoder whose patch sizes and temporal resolutions are carefully matched to each visual scale. At different granularities, it requires a corresponding multi-scale EEG encoder whose patch sizes and temporal resolutions are carefully matched to each visual scale.

## 4 Objectives

This project pursues to design and validate SUPA-EEG, a scale-unified parieto-occipital architecture that addresses key limitations of existing EEG-to-vision decoding methods. Specifically, we aim to learn soft channel selection to emphasise visually relevant occipital and parieto-occipital electrodes, extract multi-scale spatiotemporal EEG representations encoder with increasing receptive fields, and align these scale-specific embeddings to the corresponding intermediate features of a frozen I-JEPA or Transformer encoder using scale-wise InfoNCE and SIGReg stabilisation. The method will be evaluated on the THINGS-EEG dataset under the 200-way zero-shot retrieval protocol, with the goal of surpassing current state-of-the-art baselines in both Top-1 and Top-5 accuracy, while also exploring LLM-based encoding and JEPA pretraining as auxiliary validation axes.

# 5 Methodology

## 5.1 Dataset

All experiments are conducted on the THINGS-EEG dataset [4], a large-scale benchmark pairing high-density EEG recordings with natural images drawn from the THINGS object concept database [6]. The dataset comprises 10 participants, each exposed to 16,740 training image conditions repeated 4 times and 200 test image conditions repeated 80 times, yielding approximately 66,160 training trials and 16,000 test trials per subject. EEG signals were recorded using a 64-channel system at 1,000 Hz and preprocessed following the default pipeline provided by the dataset authors. Also, signals were denoised, epoched, baselined, and downsampled to 100 Hz. Then, 17 of the 64 channels are selected to keep most relevant to visual processing and were retained, resulting in a per-trial EEG tensor of shape $17 \times 100$. The visual stimuli comprise 16,940 unique object concepts from the THINGS database, partitioned into 16,740 training images and 200 test images held out for evaluation.

## 5.2 Baseline

We will evaluate SUPA-EEG against four established baselines on the THINGS-EEG benchmark [4], all of which are reported under the same 200-way zero-shot retrieval protocol. Also, table 1 summarizes the performance of each baseline across all 10 subjects under Top-1 and Top-5 accuracy from CognitionCapturer [11].

Table 1: Baseline performance on THINGS-EEG 200-way zero-shot retrieval. Top-1 (top row) and Top-5 (bottom row) accuracy are reported per subject and averaged across all 10 subjects in same participant for train/test (intra-subject)

| Method | sub-01 | sub-02 | sub-03 | sub-04 | sub-05 | sub-06 | sub-07 | sub-08 | sub-09 | sub-10 | Ave |
|---|---|---|---|---|---|---|---|---|---|---|---|
| BraVL | 6.1 | 4.9 | 5.6 | 5.0 | 4.0 | 6.0 | 6.5 | 8.8 | 4.3 | 7.0 | 5.8 |
| | 17.9 | 14.9 | 17.4 | 15.1 | 13.4 | 18.2 | 20.4 | 23.7 | 14.0 | 19.7 | 17.5 |
| NICE | 12.3 | 10.4 | 13.1 | 16.4 | 8.0 | 15.1 | 15.2 | 20.0 | 13.1 | 14.9 | 13.8 |
| | 36.6 | 33.9 | 39.0 | 47.0 | 26.9 | 40.6 | 42.1 | 49.9 | 37.1 | 41.9 | 39.5 |
| ATM | 25.6 | 22.0 | 25.0 | 31.4 | 12.9 | 21.3 | 30.5 | 38.8 | 24.4 | 29.1 | 26.1 |
| | 60.4 | 54.5 | 62.4 | 60.9 | 43.0 | 51.1 | 61.5 | 72.0 | 51.5 | 63.5 | 58.1 |
| CognitionCapturer | 27.22 | 28.72 | 37.19 | 37.69 | 21.84 | 31.55 | 32.80 | 47.60 | 33.36 | 35.07 | 33.30 |
| | 59.50 | 56.95 | 66.10 | 63.20 | 47.75 | 58.05 | 59.55 | 73.50 | 57.64 | 63.57 | 60.58 |

CognitionCapturer [11] represents the current state of the art, achieving an average Top-1 of 33.30 and Top-5 of 60.58, and serves as the primary target for SUPA-EEG to surpass. Also, the experiment will be setup with same participant for train/test (intra-subject) and difference participant for train/test (inter-subject)

## 5.3 Architecture

Figure 1 illustrates the end-to-end pipeline for training. Four stages execute in sequence.

- **Visual feature extraction:** Each feature extraction is processed by one of two encoder variants under evaluation which are A frozen I-JEPA encoder or CLIP Model processes all THINGS-EEG stimulus images once. Then all extracting features are separated into at three depth levels which are S1 capturing local edges and colours, S2 capturing spatial structure, and S3 capturing high-level object semantics. These vectors are stored as a fixed lookup table and never updated.

- **EEG tokenisation:** EEG signals are passed through an EEGNet tokeniser, producing spatiotemporal tokens with 2D positional encodings indicating channel group and time window of origin.

- **Encoding:** Each token sequence is processed by one of two encoder variants under evaluation which are a JEPA-based encoder conditioned on learned scale vectors producing scale-specific embeddings, or an LLM-based encoder treating the token sequence as a

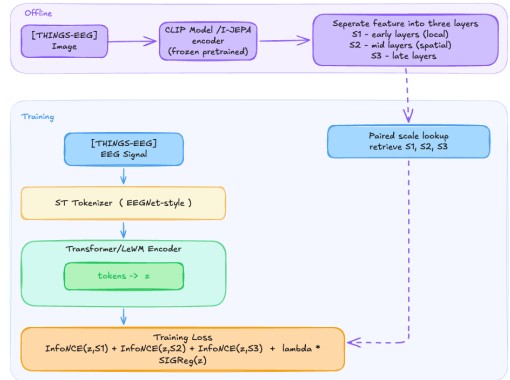

Figure 1: Model architecture for training

language-like input producing a single embedding. Both variants are evaluated under the same training objective and retrieval protocol.

- **Alignment and regularisation:** Next, $\mathbf{z}$ is aligned to its stored visual target $\mathbf{S}$ via InfoNCE and also with a SIGReg prevents representation collapse on the small ten-subject dataset. The training objective is:

$$\mathcal{L} = \sum_{k=1}^{3} \mathcal{L}_{\text{InfoNCE}}(\mathbf{z}, \mathbf{S}_k) + \lambda \cdot \mathcal{L}_{\text{SIGReg}}(\mathbf{z}) \qquad (1)$$

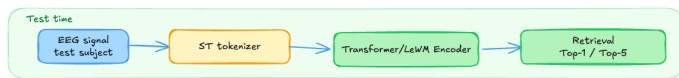

Figure 2: Model architecture for training

At test time, a EEG signal is passed through the tokeniser and the frozen learned encoder with fixed scale vectors to produce embedding data. Then, we do a retrieval via cosine similarity for 200-way zero-shot retrieval. No image encoder, feature bank, or generative model is required at inference, making our pipeline significantly leaner than reconstruction-based approaches such as CognitionCapturer [11].

## 5.4 DATA COLLECTION AND PREPROCESSING

For the pre temporal windowing and feature extraction, designed to improve signal quality, align inputs with neurophysiological processes, and reduce inter-subject variability.

**Temporal Windowing:** EEG responses to visual stimuli are temporally structured, with discriminative activity concentrated in specific post-stimulus intervals. Using the full trial introduces noise from less informative regions. We therefore explore three windowing strategies in comparison to using the full EEG sample:

- **Sliding window with overlap:** segments each trial into overlapping fixed-length windows to capture local temporal patterns while reducing boundary effects.
- **Fixed windows:** aligns windows with canonical stages of visual processing (early, mid, late), incorporating neuroscientific priors.
- **Adaptive windowing:** selects windows via change-point detection, enabling subject-specific alignment with informative activity.

**Feature Extraction :** Each window is represented either as a raw signal or through engineered features.

**Raw signal :** The original $62 \times T$ window is retained for deep models (e.g., EEGNet, LLMFew).

**Engineered features :** We extract complementary features across multiple domains:

Table 2: EEG feature categories and their functional roles in classification.

| Category | Description / Role |
|---|---|
| Time-domain | Captures temporal characteristics of EEG signals, including amplitude variations and waveform shape, which reflect event-related neural responses. |
| Frequency-domain | Represents how signal power is distributed across frequency bands, providing insight into neural oscillations linked to cognitive states. |
| Time–frequency | Combines temporal and spectral information to characterize non-stationary dynamics and transient patterns in EEG activity. |
| Spatial & clinical | Encodes spatial relationships across electrodes and clinically relevant patterns such as hemispheric asymmetry and task-related activation. |
| Higher-order | Measures complex interactions in the signal, such as phase consistency and cross-frequency coupling, reflecting functional connectivity and coordination. |

**Feature Design and Representation :** Feature extraction is treated as a configurable design space, evaluating individual groups, domain-based combinations, and full concatenation under identical settings.

**Normalization :** To ensure consistent scaling and stable optimization, features can be normalized using training-set statistics (per channel and feature).

## 6 EXPERIMENTS

Under the SUPA-EEG architecture (scale-conditioned soft channel selection, multi-scale patching, and scale-wise InfoNCE + SIGReg alignment), we adapt several encoder families into the training pipelines, for examples: a JEPA-based spatiotemporal encoder, a standard Transformer encoder and an LLM-based encoder. All variants are trained on THINGS-EEG [4] using the same multi-scale visual targets from a frozen I-JEPA encoder. The CLIP-based baselines remain unchanged and serve as offline reference benchmarks for comparison. Evaluation follows the 200-way zero-shot retrieval protocol (Top-1/Top-5 accuracy, averaged across 10 subjects, 5 random seeds). Ablations within SUPA-EEG isolate the contribution of soft channel selection, multi-scale patching (fine/medium/coarse), and the visual alignment target (I-JEPA vs. CLIP). The full model is compared against all four baselines under the same 200-way zero-shot retrieval protocol, with success defined as surpassing CognitionCapturer [11] on average Top-1 and Top-5 accuracy across all 10 subjects.

## 7 PROJECT SCHEDULE

| Week | Dates | Tasks |
|---|---|---|
| 5 | Mar 23 - 29 | Group Formation, Topic Selection |
| 6 | Mar 30 - Apr 5 | Exploring 3 Improvements to Existing EEG Methods |
| 7 | Apr 6 - 12 | From Trial and Error to Research Pipeline with a dataset |
| 8 | Apr 13 - 19 | Finalizing Research Pipeline |
| 9 | Apr 20 - 26 | Project Proposal Preparation |
| 10 | May 4 – May 10 | Implement proposal method with Transformer model |
| 11 | May 11 – May 17 | Implement proposal method with JEPA model |
| 12 | May 18 – May 24 | Experiment |
| 13 | May 25 – May 31 | Experiment and report write-up |
| 14 | Jun 1 – Jun 7 | Final paper polish, code release |

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
