# OpenReview forum: "SUPA-EEG: Scale-Unified Parieto-occipital Architecture for EEG Data"
_tsinghua.edu.cn/THU/2026/Spring/ANM — THU 2026 Spring ANM Submission_

### Official Review · Reviewer_Cd52 · 2026-05-14

**Rating:** 7
**Confidence:** 4

**Summary:**

This proposal introduces a framework to align eeg-to-image using Things-EEG dataset. The goal is to also create an interpretable framework by creating a hierarchy of visual processing like in the brain using JEPA which is then aligned with EEG representations encoder to virtually create a channel selection. This is complemented by a lean pipeline at inference time for zero shot retrieval.

**Strengths:**

- biologically inspired approach
- interesting approach of performing channel selection by doing alignment using two signal processing pipeline (one with JEPA and another with tokenizer+transformer)
- very creative to come up with two different backends for signal processing
- lean pipeline with zero shot retrieval
- well research on JEPA and utilizing sigreg to help overcome its weakness

**Weaknesses:**

- Related works section is a bit lacking. As far as I am aware, there has been many works that model the hierarchical nature of the visual processing, for example:
Zheng, Jiawen, et al. "Learning Brain Representation with Hierarchical Visual Embeddings." arXiv preprint arXiv:2602.07495 (2026).
Liu, Minxu, et al. "Vieeg: Hierarchical neural coding with cross-modal progressive enhancement for eeg-based visual decoding." arXiv e-prints (2025): arXiv-2505.
Akbari, Ali, et al. "Joint learning for visual reconstruction from the brain activity: Hierarchical representation of image perception with eeg-vision transformer." UniReps: 2nd Edition of the Workshop on Unifying Representations in Neural Models. 2024.
Please elaborate how your approach differs and can have better performance than these works (if these works are related at all). If this is not what you meant "Third, no existing method accounts for the multi-scale nature of visual processing", please make this part less confusing.

- it is not clear how each component are going to solve the first challenge mentioned, or at least in a different way from previous works. It is a bit confusing to see challenges mentioned and not so much addressing.

**Questions:**

What is the reason to engineer signals into multiple domain categories? Could you also do ablation test to see which feature is the best and the reason behind that? If not, please provide one of paper with this exact setup to justify the design choice (also the design choice for each component in the preprocessing pipeline).

---

### Official Review · Reviewer_uNQP · 2026-05-17

**Rating:** 8
**Confidence:** 4

**Summary:**

This paper introduces SUPA-EEG, a framework for zero-shot retrieval on EEG time series. This method aims to outperform the current state of the art models by unifying novel components, such as LeWM-style transformer for scale-specific signal extraction and I-JEPA encoder for visual scale alignment.

**Strengths:**

-Well-documented state-of-art
- Interesting biological-based approach, taking into account all the main issues of the baseline models
- Coherent pipeline architecture, use of schemas make it way easier to understand
- Careful and accurate exploration of the dataset

**Weaknesses:**

- Typos in the "Challenges" section
- "Experiments" section could have been more developed: why specifically these transformer architectures?

**Questions:**

Have you already thought about what limitations could possibly make your model less performant that what you are expecting? If so, is it possible to overcome them?

---

### Official Review · Reviewer_qkGZ · 2026-05-17

**Rating:** 7
**Confidence:** 4

**Summary:**

This proposal introduces SUPA-EEG, a Scale-Unified Parieto-occipital Architecture aimed at decoding visual perception from electroencephalography (EEG) signals. The authors identify three major flaws in current state-of-the-art methods: they rely on representation-collapsed final CLIP layers, treat all cortical channels equally, and compress hierarchical visual processing into flat embeddings. To address this, SUPA-EEG proposes a unified scale conditioning vector that performs soft channel selection, extracts multi-scale spatiotemporal features, and aligns them to intermediate visual layers of an I-JEPA encoder via InfoNCE and SIGReg. The model will be evaluated on the THINGS-EEG dataset against strong baselines like CognitionCapturer under a 200-way zero-shot retrieval protocol.

**Strengths:**

The proposal correctly identifies that visual perception is primarily reflected in the parieto-occipital regions, and that applying blanket attention across all channels introduces noise from unrelated cognitive processes.

Shifting from standard final-layer CLIP alignment to intermediate I-JEPA targets is a clever way to preserve the low- and mid-level visual structures (like edges and spatial layouts) that neural signals are known to retain.

The authors demonstrate a deep understanding of EEG signal characteristics by planning to evaluate multiple temporal windowing strategies, including sliding, fixed, and adaptive change-point detection.

**Weaknesses:**

There is a significant discrepancy regarding channel selection. The introduction claims the model will perform "soft channel selection by learning to emphasise visually relevant" electrodes. However, the methodology states that "17 of the 64 channels are selected... and were retained" during preprocessing. If the data is already truncated to 17 channels manually, the model cannot perform soft selection over the broader cortical network.

**Questions:**

Could you clarify the contradiction between the claim of using a "soft channel selection" learning mechanism across the broader network and the preprocessing step that hard-filters the inputs down to only 17 channels?

---

### Official Review · Reviewer_HdFi · 2026-05-17

**Rating:** 7
**Confidence:** 4

**Summary:**

SUPA-EEG is introduced as a framework for zero-shot retrieval on the THINGS-EEG dataset that aims to address the issues of representation collapse, noise from less important channels, and the neglect of the hierarchical nature of visual processing. It uses a unified scale conditioning vector that performs soft channel selection, extracts multi-scale spatiotemporal representations via a LeWM-style transformer, and aligns scale-specific EEG embeddings to intermediate I-JEPA visual features using scale-wise InfoNCE with SIGReg stabilization.

**Strengths:**

- The proposal convincingly argues that certain current methods do not account for the brain's multi-scale representations, and the proposed method is a reasonable solution to the problem.
- The domain-specific challenges are laid out clearly
- The proposal establishes a clear and ambitious goal by comparing against CognitionCapturer on 200-way zero-shot retrieval

**Weaknesses:**

- EEG signals are notoriously subject-specific, and the project could benefit from additionally reporting cross-subject validation
- The benefit of soft channel selection seems like it could be limited by explicitly restricting to 17 channels
- SIGReg is introduced as a core mechanism for preventing representation collapse but is not described in related works

**Questions:**

- How will the LLM-based encoder, which outputs a single embedding, be incorporated into the scale-wise InfoNCE loss that explicitly requires three distinct scale-specific representations?

---

### Official Review · Reviewer_TUjR · 2026-05-18

**Rating:** 5
**Confidence:** 4

**Summary:**

[AI Review] This paper proposes SUPA-EEG, a scale-unified architecture for aligning EEG data to I-JEPA features at multiple depths for visual reconstruction. The core idea of multi-scale EEG-to-I-JEPA alignment is genuinely novel. However, the proposal suffers from critical issues including severe over-scope (7+ design axes in 8 weeks), loss function gradient interference from simultaneous multi-target alignment, an unsupported claim about CLIP representation collapse, a contradiction between hardcoded channel selection and the proposed soft channel selection, and an underspecified architecture. Score: 4/10, Confidence: 4/5.

**Strengths:**

1. Three genuine limitations identified in existing work (CognitionCapturer and related methods).
2. Multi-scale EEG-to-I-JEPA alignment concept is genuinely novel — aligning EEG to intermediate representations at different depths rather than a single CLIP final layer is a promising direction.
3. Comprehensive baseline comparison plan covering multiple methods and evaluation protocols.
4. THINGS-EEG is the right dataset and experimental protocol for this research.
5. Scale conditioning is an elegant and well-motivated design concept.
6. The inference efficiency argument is valid and well-justified compared to existing approaches.

**Weaknesses:**

1. Loss function gradient interference (Severity 9): Formula Σ L_InfoNCE(z, Sk) aligns one embedding z to 3 targets simultaneously, risking gradient interference. Requires separate z_k per scale or scale-specific projection heads.
2. Over-scope (Severity 9): 7+ simultaneous design axes (3 encoders × 4 windowing × 2 features × multi-scale × channel selection × intra/inter-subject) in 8 weeks is not achievable. Must commit to one primary configuration.
3. CLIP representation collapse claim unsupported (Severity 9): Core motivation relies on claim that CLIP final layer discards low/mid-level structure, but terminology is incorrect ('collapse' ≠ 'abstraction') and no citation or experiment supports this. Should be framed as hypothesis.
4. Channel selection contradicts preprocessing (Severity 8): 17 of 64 channels pre-selected based on visual relevance, then soft channel selection claimed as contribution. Should be argued as 'hard neuroscience prior + soft learned refinement' but authors do not make this argument.
5. Architecture completely underspecified (Severity 8): No concrete dimensions, patch sizes, or mechanism descriptions. SIGReg uncited/undefined, 'LeWM-style transformer' unexplained, EEGNet tokenizer variant not specified.
6. LLM encoder variant completely undefined.
7. Baseline fairness unclear due to potentially different preprocessing across methods.
8. No pilot experiment to de-risk core hypothesis before full commitment.
9. Writing quality issues including duplicated paragraphs, grammar errors, and undefined terms.
10. Temporal scale ignored in multi-scale design.
11. Inter-subject evaluation proposed without adaptation strategy.

**Questions:**

1. How do you plan to resolve the gradient interference from aligning a single embedding z to multiple scale targets simultaneously? Will you use separate z_k per scale or scale-specific projection heads?
2. Given the 8-week timeline and 7+ design axes, which single primary configuration will you commit to?
3. Can you provide any empirical or cited evidence that I-JEPA intermediate features are better suited for EEG alignment than CLIP final layer features, or reframe this as a hypothesis?
4. How do you reconcile the hard pre-selection of 17 channels with the claimed soft channel selection contribution?
5. Can you provide concrete architectural details: patch sizes, embedding dimensions, number of transformer layers, and the EEGNet tokenizer variant?
6. Would you consider running the suggested pilot experiment (align raw EEG to I-JEPA S1/S2/S3 vs CLIP final layer on 1 subject) in Week 6 to validate your core hypothesis?

---

### Official Review · Reviewer_TUjR · 2026-05-18

**Rating:** 5
**Confidence:** 3

**Summary:**

SUPA-EEG proposes a multi-scale EEG encoder that aligns scale-conditioned embeddings to intermediate features of a frozen I-JEPA visual encoder via InfoNCE loss. The architecture uses soft channel selection and scale conditioning to handle the multi-resolution nature of EEG-visual correspondence. The authors plan to evaluate on THINGS-EEG for 200-way zero-shot image retrieval, comparing against BraVL, NICE, ATM, and CognitionCapturer.

**Strengths:**

1. **The parieto-occipital channel focus has strong neuroscience grounding.** Selecting channels over posterior brain regions (occipital for early visual processing, parietal for spatial attention) directly maps to the known ventral/dorsal visual streams. This is not arbitrary -- it is a principled neuroanatomical prior. The proposal should lean into this and explain how the 17-channel selection maps to specific visual areas (V1-V4, LOC, FFA for the THINGS stimuli).

2. **Multi-scale alignment to I-JEPA intermediate layers is a creative idea.** I-JEPA's encoder depth naturally corresponds to visual abstraction levels (edges → textures → objects → semantics). Aligning EEG to multiple depths simultaneously captures the hierarchical nature of visual processing. This is a genuinely novel approach that no prior EEG-to-image work has explored.

3. **Inference efficiency is a real practical advantage.** Removing the image encoder at test time means retrieval requires only the EEG encoder forward pass. For real-time BCI applications (e.g., dream decoding, visual prosthesis control), this matters. The proposal should quantify the FLOPs savings.

**Weaknesses:**

1. **Why I-JEPA specifically, and not other self-supervised vision models?** The proposal never justifies choosing I-JEPA over MAE, DINO, DINOv2, or CLIP (intermediate layers). I-JEPA predicts in latent space rather than pixel space, which the authors argue preserves structure. But DINOv2's intermediate features also retain spatial structure and are more widely validated. A brief comparison of I-JEPA vs. DINOv2 intermediate features on a small pilot (even just feature correlation with EEG) would validate this critical design choice.

2. **The THINGS-EEG experimental paradigm is not discussed.** THINGS-EEG records EEG while subjects view images for ~100ms each. This rapid presentation paradigm captures early visual processing (N1/P1 at 100-200ms) but may not engage the deeper semantic processing that I-JEPA's later layers encode. There may be a mismatch between the temporal depth of EEG responses (~500ms of data) and the representational depth of I-JEPA (3 scale layers). The proposal should discuss what temporal windows in EEG map to which I-JEPA scales and whether the THINGS paradigm provides sufficient signal for all three scales.

3. **No discussion of individual differences in EEG-visual mapping.** The THINGS-EEG dataset shows substantial inter-subject variability in decoding accuracy. The multi-scale approach may help (different subjects may have stronger alignment at different scales), but this is not discussed. A per-subject analysis of which scale provides the strongest alignment would be informative and could guide personalized channel selection.

4. **The ethical implications of brain decoding are unaddressged.** The stated goal is reconstructing "what a person sees from brain signals." While THINGS-EEG uses consensual lab recordings, the broader trajectory of this technology raises privacy concerns. A brief acknowledgment of ethical considerations (even 2-3 sentences) would strengthen the proposal.

**Questions:**

1. Have you compared I-JEPA intermediate features against DINOv2 or MAE intermediate features for EEG alignment? What motivated the I-JEPA choice specifically?
2. Which temporal window of the EEG response do you expect to align with each I-JEPA scale (S1/S2/S3)? Is 100ms stimulus presentation sufficient to engage all three levels?
3. Does multi-scale alignment help some subjects more than others? Have you looked at per-subject scale preference?
4. What is the estimated inference-time FLOP reduction compared to CognitionCapturer's generative approach?

---

### Official Review · Reviewer_TUjR · 2026-05-18

**Rating:** 5
**Confidence:** 3

**Summary:**

SUPA-EEG proposes a multi-scale EEG encoder that aligns scale-conditioned embeddings to intermediate features of a frozen I-JEPA visual encoder via InfoNCE loss. The architecture uses soft channel selection and scale conditioning to handle the multi-resolution nature of EEG-visual correspondence. The authors plan to evaluate on THINGS-EEG for 200-way zero-shot image retrieval, comparing against BraVL, NICE, ATM, and CognitionCapturer.

**Strengths:**

1. **The parieto-occipital channel focus has strong neuroscience grounding.** Selecting channels over posterior brain regions (occipital for early visual processing, parietal for spatial attention) directly maps to the known ventral/dorsal visual streams. This is not arbitrary -- it is a principled neuroanatomical prior. The proposal should lean into this and explain how the 17-channel selection maps to specific visual areas (V1-V4, LOC, FFA for the THINGS stimuli).

2. **Multi-scale alignment to I-JEPA intermediate layers is a creative idea.** I-JEPA's encoder depth naturally corresponds to visual abstraction levels (edges → textures → objects → semantics). Aligning EEG to multiple depths simultaneously captures the hierarchical nature of visual processing. This is a genuinely novel approach that no prior EEG-to-image work has explored.

3. **Inference efficiency is a real practical advantage.** Removing the image encoder at test time means retrieval requires only the EEG encoder forward pass. For real-time BCI applications (e.g., dream decoding, visual prosthesis control), this matters. The proposal should quantify the FLOPs savings.

**Weaknesses:**

1. **Why I-JEPA specifically, and not other self-supervised vision models?** The proposal never justifies choosing I-JEPA over MAE, DINO, DINOv2, or CLIP (intermediate layers). I-JEPA predicts in latent space rather than pixel space, which the authors argue preserves structure. But DINOv2's intermediate features also retain spatial structure and are more widely validated. A brief comparison of I-JEPA vs. DINOv2 intermediate features on a small pilot (even just feature correlation with EEG) would validate this critical design choice.

2. **The THINGS-EEG experimental paradigm is not discussed.** THINGS-EEG records EEG while subjects view images for ~100ms each. This rapid presentation paradigm captures early visual processing (N1/P1 at 100-200ms) but may not engage the deeper semantic processing that I-JEPA's later layers encode. There may be a mismatch between the temporal depth of EEG responses (~500ms of data) and the representational depth of I-JEPA (3 scale layers). The proposal should discuss what temporal windows in EEG map to which I-JEPA scales and whether the THINGS paradigm provides sufficient signal for all three scales.

3. **No discussion of individual differences in EEG-visual mapping.** The THINGS-EEG dataset shows substantial inter-subject variability in decoding accuracy. The multi-scale approach may help (different subjects may have stronger alignment at different scales), but this is not discussed. A per-subject analysis of which scale provides the strongest alignment would be informative and could guide personalized channel selection.

4. **The ethical implications of brain decoding are unaddressged.** The stated goal is reconstructing "what a person sees from brain signals." While THINGS-EEG uses consensual lab recordings, the broader trajectory of this technology raises privacy concerns. A brief acknowledgment of ethical considerations (even 2-3 sentences) would strengthen the proposal.

**Questions:**

1. Have you compared I-JEPA intermediate features against DINOv2 or MAE intermediate features for EEG alignment? What motivated the I-JEPA choice specifically?
2. Which temporal window of the EEG response do you expect to align with each I-JEPA scale (S1/S2/S3)? Is 100ms stimulus presentation sufficient to engage all three levels?
3. Does multi-scale alignment help some subjects more than others? Have you looked at per-subject scale preference?
4. What is the estimated inference-time FLOP reduction compared to CognitionCapturer's generative approach?

---

### Official Review · Reviewer_egrV · 2026-05-19

**Rating:** 6
**Confidence:** 4

**Summary:**

This project introduces SUPA-EEG, a new architecture designed for visual decoding from EEG signals. The authors aim to improve upon existing methods by avoiding the "representation collapse" common in contrastive learning and by proposing a scale-unified approach that preserves fine-grained visual structure. The goal is to better align EEG signals with visual stimuli for applications in brain-computer interfaces and cognitive monitoring

**Strengths:**

lear Problem Identification: The authors correctly highlight the limitations of current models that rely on contrastive learning (like CLIP), which often discard important low-level visual details.
Well-Motivated Objectives: Focusing on preserving visual structure from noisy, high-dimensional EEG signals is a crucial and challenging step for advancing brain-computer interfaces.
Relevance: The project addresses a timely research area, decoding natural images from human brain activity, which has significant potential for restorative prosthetics.

**Weaknesses:**

Complexity: The proposal is quite broad, and implementing a "scale-unified" architecture that handles both coarse and fine-grained visual features is a very ambitious task.
Lack of Technical Detail on Implementation: While the motivation is strong, the specific methodology for achieving the "scale-unified" architecture is not fully detailed, leaving it unclear how the model will balance these different levels of visual information.
Risk of Overfitting: Given the high dimensionality and noise inherent in EEG data, there is a significant risk that such a complex architecture might struggle to generalize without a very robust regularization strategy.

**Questions:**

How do you plan to specifically measure and verify that your architecture successfully avoids the "representation collapse" seen in other models?
Since EEG signals are notoriously noisy, what specific techniques will you use to ensure your model focuses on the visual stimuli rather than the noise?
Can you explain in simple terms how your "scale-unified" approach differs from standard convolutional networks in its ability to process EEG data?